# Basic Empathy Scale: A Systematic Review and Reliability Generalization Meta-Analysis

**DOI:** 10.3390/healthcare10010029

**Published:** 2021-12-24

**Authors:** Javier Cabedo-Peris, Manuel Martí-Vilar, César Merino-Soto, Mafalda Ortiz-Morán

**Affiliations:** 1Department of Basic Psychology, Faculty of Psychology and Speech Therapy, Universitat de València, 46010 Valencia, Spain; javi.cabe.peris@gmail.com; 2Research Institute of the School of Psychology, Universidad de San Martín de Porres, Lima 15102, Peru; 3Department of Psychology, Faculty of Psychology, Universidad Nacional Federico Villarreal, Lima 15088, Peru; mortiz@unfv.edu.pe

**Keywords:** prosocial behavior, empathy, meta-analysis, reliability, systematic review

## Abstract

The Basic Empathy Scale (BES) has been internationally used to measure empathy. A systematic review including 74 articles that implement the instrument since its development in 2006 was carried out. Moreover, an evidence validity analysis and a reliability generalization meta-analysis were performed to examine if the scale presented the appropriate values to justify its application. Results from the systematic review showed that the use of the BES is increasing, although the research areas in which it is being implemented are currently being broadened. The validity analyses indicated that both the type of factor analysis and reliability are reported in validation studies much more than the consequences of testing are. Regarding the meta-analysis results, the mean of Cronbach’s *α* for cognitive empathy was 0.81 (95% CI: 0.77–0.85), with high levels of heterogeneity (*I*^2^ = 98.81%). Regarding affective empathy, the mean of Cronbach’s *α* was 0.81 (95% CI: 0.76–0.84), with high levels of heterogeneity. It was concluded that BES is appropriate to be used in general population groups, although not recommended for clinical diagnosis; and there is a moderate to high heterogeneity in the mean of Cronbach’s *α*. The practical implications of the results in mean estimation and heterogeneity are discussed.

## 1. Introduction

Empathy has been defined in a multitude of ways by different authors since Titchener coined the term in 1909 [1]. Eklund and Meranius [2] conducted a review of their conceptualization, and found 52 documents of which common themes were grouped into 13 subcategories. The predominant commonality in all of them was focused on four attributes: understanding, feeling, and sharing emotions; and the emotional differentiation with others. These four attributes form two key components of the conceptual structure of empathy: the cognitive and affective dimensions. The present study is regarded as highly relevant as it can provide an in-depth analysis of the Basic Empathy Scale (BES) [3], as well as a qualitative and quantitative study of its validity and reliability. Reviews and meta-analyses are not commonly found in the scientific literature; however, the authors of this study consider them important to help researchers and professionals.

### 1.1. The Development of the BES

Jolliffe and Farrington [3] designed a self-report measure with the aim of overcoming the flaws in the scales that were being used to measure empathy until that point in time. The Interpersonal Reactivity Index [4], the Hogan Empathy Scale [5], and the Questionnaire Measure of Emotional Empathy [6] were some examples of such scales. These deficiencies were mainly: (a) the lack of sensitivity towards the differentiation between empathy and sympathy, both conceptually and psychometrically; (b) the role of cognitive empathy; and (c) the lack of representative samples of the general population in their validation.

Jolliffe and Farrington [3] briefly described the theoretical and pragmatic consequences of this lack of both discriminatory validity and sample representativeness. On the one hand, the lack of differentiation between sympathy and affective empathy can produce difficulties in the assignment of the objectives of intervention in a population lacking one of them. This is because the first one (i.e., empathy) does not always have to be congruent with the described emotion, whereas the second one (i.e., sympathy) does [7,8]. Regarding the conceptualization of cognitive empathy, it is necessary to understand the contribution of empathy to different situations, such as crime. Someone can understand the feelings of the victim (cognitive empathy), while showing no emotional reaction to such feelings; that is, without feeling what the victim feels (low affective empathy). This situation can cause someone to exert certain actions on their victim that are not aimed at alleviating suffering. Finally, regarding the lack of sample representativeness, most of the studies were conducted on university students, who usually share characteristics that cannot be extrapolated to the general population, or to other randomly extracted samples. Although student samples can be efficiently used to help with the measurement design [9], there is sufficient scientific literature which makes robust claims about the poor generalization of student samples [10], applied to several psychosocial constructs, such as empathy.

In response to these challenges, the BES was created, a self-reported scale focused on four of the five basic emotions: fear, sadness, anger, and happiness [3]. For this instrument, the definition of empathy used was based on the one provided by Cohen and Strayer [11]. This definition was considered the most appropriate due to the importance it gives to both the affective and cognitive aspects of empathy, as it understands that they are two differentiated constructs within the same variable.

### 1.2. Description of the Scale

The BES [3] consists of 20 items, which are divided into two factors: cognitive empathy (9 items) and affective empathy (11 items). Each item has five ordinal response alternatives, ranging from 1 (no agreement) to 5 (full agreement). The authors highlight that the BES is a multidimensional scale, because it clearly distinguishes cognitive from affective empathy. They also indicate that a total empathy score can be obtained by adding the value of all items, so that the total value can range between 20 (low empathy) and 100 (high empathy) [3,12]. The scale obtained good evidence of construct validity and internal consistency in its first development, and it presented Cronbach’s *α* values of 0.79 for the cognitive empathy subscale, and 0.85 for the affective empathy subscale. Confirmatory factor analysis showed the existence of two scales, although they correlated with statistical significance. No positive correlations were found between empathy and social desirability [3].

During the validation of the instrument, the authors used a sample composed of adolescents [3,13,14]. The reason for choosing this specific sample was that they showed the highest relationship between a lack of empathy and two of the most measured topics with the BES instrument: crime and bullying. These authors noted that those people with low emotional empathy were more likely to engage in occasional and frequent bullying episodes [13]. Jolliffe and Farrington [12] conducted an updated review of the implementation of the BES to examine the relationship between empathy and parenthood, antisocial behavior, harassment, and delinquency in studies composed of either English-speaking populations or their international adaptations [15,16,17,18,19,20]. Although these have been the most addressed topics, the instrument is also currently being used for the general population.

The BES has been globally administered in Chinese, French, Italian, Portuguese, Slovak, Peruvian, and Spanish adaptations [12,21]. In addition, an abbreviated version was created in Spain, the Basic Empathy Scale—Brief version (BES-B), which only contains nine items [22]. This adaptation is also composed of the factors of affective empathy (four items) and cognitive empathy (five items). Its validation in Peru showed the expected dimensionality (two dimensions), a reliability higher than 0.70 in children and adolescents [23,24], and measurement invariance. 

Although, in principle, the only factors it measures are affective and cognitive empathy, various international studies have tried to observe the obtained effects when the empathy variable is divided into three subscales: emotional contagion and emotional disconnection, resulting from the division of the affective empathy factor; and cognitive empathy [25,26,27]. These three subscales derive from the three elements in the construction of empathy according to the neurodevelopmental model based on the stages of childhood proposed by Decety and Svetlova [28]. This is composed of empathic understanding, which would correspond to cognitive empathy; empathic concern, which would relate to emotional disconnection; and affective activation, in line with emotional contagion. This differentiation between two or three subscales can be decisive when calculating the reliability of the instrument, due to the distribution of items. The fact that the instrument has been initially developed based on two factors should be decisive when choosing the most appropriate factor configuration, since choosing the three-subscale model would likely require an adaptation.

### 1.3. Proposals for the Study of the Scale Properties

The BES is one of the most widely used instruments to measure empathy throughout the world. The first article in which it was described and validated by its authors has been cited more than 850 times, with a progressive increase (according to a search in Google Scholar that the authors themselves carried out in 2018) [12]. However, the meta-analytic evaluation of the psychometric properties of the BES seems to be absent. After performing an initial search in the Web of Science (WoS), Scopus, PsycInfo, and Dialnet databases, no previous meta-analyses on the psychometric properties of this instrument were found. However, it should be noted that in this research, some systematic review studies of other empathy instruments were found, where the reliability and validity of the instruments are determined [29], as well as the purpose of analyzing and assessing the rigor of empathy measures [30]. Studies specifically dedicated to the analysis of the evidence of validity, within its multiple forms, were also not found, nor studies that assessed the reliability of the instrument. The description of the evidence of validity regarding the empathy construct requires a consensual framework to identify the force of validity of the scores of an instrument. At the same time, this description can be useful for conceptualizing studies of measurement validity. To date, measurement standards are an internationally accepted proposal for this purpose [31].

The American Educational Research Association (AERA), in collaboration with the American Psychological Association (APA) and the National Council on Measurement in Education (NCME), created a guide to study the validity of instruments, which is usually called “Standards” [31]. It presents guidelines for assessing the composition, use, and interpretation of what a test is intended to measure. This guide explains that validity depends on specific interpretations, and is aimed for particular uses of the scores of a measure, and, therefore, it is incorrect to believe that there is a general validity of a test. The “Standards” propose five sources in which validity can be tested: test content, response processes, internal structure, relations to other variables, and consequences of testing. The validity of content, response processes, and test consequences are interpreted predominantly by qualitative methods, whereas the validity of internal structure and relationship with other variables are done so predominantly using quantitative methods.

The definitions of each source of validity are reflected in “Standards” [25], and are summarized below. The test content validity is responsible for corroborating that the instruments measure the construct they should measure. The validity of response processes observes the analysis that studies carry out about how participants have interpreted the items when responding to them. The validity of internal structure measures the interrelations of items and constructs. The validity of relations to other variables is focused on the study of a questionnaire in its comparison with external variables to demonstrate its associative capacity with other theoretically relevant variables. Finally, the validity of the consequences of testing affects the interpretation of the effects that the test might have, which may or may not be in line with its initial intention. Due to its ability to integrate the study of validity’s characteristics from a qualitative and quantitative point of view, in addition to its evidence-based method, the “Standards” are considered an appropriate approach to carry out reviews on the validity of questionnaires [32]. Although “Standards” provide a useful and effective reference for organizing the evidences of a scale, few instruments have been reviewed with this framework to date [32], likely due to its disregard or its lack of expression in a protocol for systematic description. A registration protocol of this type has recently been reported [26], and it can be an important resource for systematic descriptive reviews. In this particular study, this protocol served as a guide for the systematic review of the BES.

The study of the reliability of the instrument is necessary, considering that it is a psychometric property that changes according to external variables per se, such as the composition of the sample or the context in which it is applied (criteria that are not always considered by researchers). Estimations of test scores should be based on their own data, rather than being induced from previous applications [33]. The performance of a reliability meta-analysis is a methodology that allows the finding of factors involved in such variability. It will also help study the adequacy and generalization of an instrument within a framework. The concept of generalization of reliability corresponds to the psychometric analysis of the properties of a questionnaire in the different situations in which it has been applied [34], being considered the only type of meta-analysis in which the main effect size indices are reliability coefficients reported in previous studies [35]. In order to operationalize and standardize the measurement of reliability generalization, a guide of the reliability and generalization of meta-analyses called “Reliability Generalization Meta-Analysis” (REGEMA) was created [35].

### 1.4. The Current Study

The main objective is the synthesis of the properties of the BES instrument throughout its international implementation, since its creation in 2006 until the present time. The study was performed using three procedures: first, a brief systematic review, which will provide an overview of the instrument; second, an analysis of the validity evidence that has been shown by those studies dedicated to the analysis of its psychometric properties, which will help to assess the quality of its content and implementation; finally, a meta-analysis on the reliability generalization, which will seek to study the internal consistency of the BES. 

In this way, we intended to obtain an overview of the studies that have made use of this questionnaire in order to know if there is a good justification for its current high application when measuring empathy. In addition, these procedures will make it possible to assess the adequacy of the instrument according to the framework of use to which it is addressed. As a final purpose, the present study tries to provide evidence of the reliability of the instrument, allowing to make decisions when using it, both in the areas of evaluation and in research. Novel contributions to the study of reliability are intended to be presented during this process. The Research Question (RQ) and the Hypothesis (H) are formulated as follows:

RQ: Does the BES provide enough levels of validity and reliability to ensure its appropriate use?

H: Due to the evidence of the BES’ widespread use, we hypothesized it would present good levels of validity and internal consistency for its use in the general population.

## 2. Materials and Methods

The procedure was divided into three steps. First, a systematic review was carried out using the PRISMA method [36]. Second, a systematic descriptive literature review of the validity evidence was conducted in accordance with the “Standards” framework [31]. Third, a meta-analysis was conducted, following the recommendations of the REGEMA checklist [35]. This manuscript was developed following the REGEMA guidelines, pursuing the demonstration of good quality in its implementation.

### 2.1. Systematic Review

The literature search was made in two iterations.

#### 2.1.1. Search Strategy and Information Sources

Two search iterations were carried out. The first iteration was conducted in WoS, Scopus, PsycInfo, and Dialnet databases. A search profile able to cover as much of the appropriate work as possible was created. Therefore, the term “Basic Empathy Scale” was introduced in the basic search mode for each database. This was an attempt to take into account all the works that had used the instrument. The results were refined to include all those articles from 2006 to 2020 (both included). English and Spanish were chosen as the inclusion language criteria. In the case of PsycInfo, the criterion “human species only” was also introduced, since it was the only one that had this tool.

The search was restricted by limiting the area to that of psychology. In the Scopus database, the fields “psychology” and “social sciences” were chosen as subject area limiters. Regarding WoS, the research area “social sciences” was chosen. In Dialnet, the domains to which the research was limited were “psychology” and “social sciences”. This step was not necessary in PsycInfo because it is a database specialized in this area.

The second iteration of search was conducted manually. The references found in the articles of the first iteration were reviewed, aiming to identify studies that met the established search criteria and had not been found in the first iteration. 

#### 2.1.2. Eligibility Criteria and Selection Process

Among the articles selected to assess their eligibility, a screening was performed to select which of them met the inclusion criteria: (a) researches in which the BES was applied in its original or reduced version; (b) researches in which the BES was applied in its original language (English) or in any other language to which it had been adapted; (c) experimental or quasi-experimental studies; (d) correlational studies with methodological focus; and (e) any target population, regardless of their age or clinical characteristics. It was not considered appropriate to introduce limitations on the age of the population that form the samples because, although the instrument was initially validated in adolescents, its implementation has been carried out in children and adults. Furthermore, the inclusion of both experimental and correlational studies helped the researchers to have a wide overview of the BES. Although it seems that this could act as a distractor from the main focus of this review, it provided the necessary reports to conduct the further analysis of validity, and the reliability meta-analysis. Regarding the exclusion criteria, the following were proposed: (a) systematic or bibliometric reviews of the instrument, (b) single case studies, and (c) studies that were not published between the established dates (2006 to 2020, both included).

The articles were screened following this eligibility criteria. This was performed two times by two different researchers. Then, their results were discussed until they arrived at a consensus. Finally, a third researcher confirmed the quality of the results. 

#### 2.1.3. Data Collection Process

First, the information from the 74 selected articles was extracted. To do this, an Excel table was filled in with the most relevant information from each section of a scientific paper: the introduction, the objectives, the hypotheses, the sample details, the results, and the discussion; as well as possible annotations on the limitations, practical implications, and future research.

Secondly, two Excel tables were created to transcribe the information related to the internal consistency of each study. In the first table, the reliability values were recorded manually in the same way they were expressed in the selected articles. The information was divided according to whether the study analyzed the Cronbach’s *α*, the McDonald’s *ω*, or a test-retest. Each value was divided according to whether it was measured for total empathy, cognitive empathy, or affective empathy. One column was left to record whether any study had performed the reliability measure for three subscales (emotional disconnection, emotional contagion, and cognitive empathy) rather than the traditional subdivision into two (cognitive empathy and affective empathy), and another column to note if the reliability had not been measured in that study. 

Finally, the second of the tables focused on internal consistency was aimed to categorize the 74 articles according to whether they included the reliability values in a reported or induced way. They were considered reported when they were calculated for the study, and induced when they did not indicate so. Within the reported category, the results were subdivided into two groups: usable and unusable. The usable ones were those that calculated reliability for at least each subscale of the instrument, and the unusable ones were those that only assessed reliability as a measure of total empathy. A separate column, not exclusive to the other categories, was added and named “not relevant”, in which articles containing a value for total empathy were included, regardless of whether they also expressed reliability in a usable form. Induced reliability was subdivided into three categories: omitted, vague, and precise. For this classification, the work of Rubio Aparicio et al. [37] was taken as an example.

Induced reliability was considered omitted when nothing about it was indicated in the study, vague when “good” reliability was expressed by citing previous studies, and precise when the exact value of another previous study was reported. All articles were divided depending on the version of the BES instrument they used, the language in which it was presented, and the number of items with which their reliability was analyzed. This table was divided into three parts, each of which (33%) was revised by a different researcher as a means of quality control. Disparities between researchers were estimated following a qualitative procedure. As a result, they were considered minor and easy to overcome. 

A linear regression was calculated to observe the publication progression of articles that used the BES instrument over time. This calculation was carried out with the 1.6 version of the *jamovi* software for statistical analysis [38].

### 2.2. Description of the Validity Study

A table exclusively containing articles that measured the psychometric properties of the instrument focusing on its validation, regardless of the language and the version of the instrument, was made. Only 21 of the 74 selected articles met these characteristics. In this table, the information related to authorship, year of publication, and each of the five standards proposed in the “Standards” [31,32] guide was extracted. Each article was labeled according to whether the validity tests presented in this guideline were, in any way, satisfied by the categorical system of “yes”, “no”, or “ambiguous”. The information taken into account to perform this analysis was extracted directly from the texts. No inferences were made by the researchers of this present study in the process.

The third of the standards, concerning the validity of the internal structure, was divided into five sections. These were the type of factor analysis carried out (exploratory, confirmatory, etc.), the way in which reliability was expressed (McDonald’s *ω* or Cronbach’s *α*), whether or not a test-retest study of reliability was presented, the study of factor invariance according to the groups in which it was divided (for example, gender) and the level of study up to which it was reached (metric, intercepts, etc.), and whether or not it had a statistical study for the analysis of equivalence between versions. This was done due to the fact that solely reporting validity evidences based on the internal structure was considered insufficient because it is a broad construct, and can offer much more useful information to justify its applicability. Finally, a review was carried out among researchers in a telematic way until a consensus for each response was reached.

### 2.3. Meta-Analysis

#### 2.3.1. Article Eligibility

To carry out the meta-analysis, only those articles that presented a reported reliability for at least the scales of affective and cognitive empathy measured with the coefficient of Cronbach’s *α* were chosen. Among them, only those which used the original 20-item scale in their original or translated version were selected. The total number of articles included was initially 31, but 2 of them reported the same reliability values because they had used the same sample. Data were extracted from the oldest of the 2, and the final number of articles was 30. In this way, we tried to homogenize the results to avoid biases during the process. All the information related to these 30 articles was included in a table that showed the data from each study related to: authors, year of publication, number of items for each factor of the instrument (affective and cognitive empathy), language of the instrument, number of subjects in the sample, sex of the sample (divided into women, men, and mixed), type of sample (divided into general or special, if it was clinical or forensic, respectively), generational group of the sample (divided into adolescents, youth, and adults), and the value of the Cronbach’s *α* coefficient for each factor (affective and cognitive empathy). The authors of this study assessed the data of this table qualitatively until reaching conformity, as was the case in the systematic review and the validity analysis. Version 3.0-2 of the R *metafor* package was used [39]. All commands used in this process can be obtained by consulting the main author.

#### 2.3.2. Description and Assessment of Cronbach’s α Coefficients

The means of Cronbach’s *α* from each study and their relative values according to their confidence interval (95%) were compared with two null values proposed beforehand [40]. These null values were established at the Cronbach’s *α* points: >0.70 y > 0.80, as they are the values that are usually considered appropriate for a reliability coefficient [41,42]. This procedure was done in order to assess the suitability of the reliability of each study.

#### 2.3.3. Reliability Generalization

Potential publication biases were assessed using the Egger’s Test [43] and the Rank’s Test [44]. Both approaches, parametric and non-parametric, respectively, were used as a means to corroborate the results reciprocally. The Eggers’s Test was assessed with the mixed effects model, with a cut-off point located on *p* = 0.20 in order to increase its sensitivity [45]. On the other hand, as the Rank’s Test is not sensitive to non-severe biases [46,47], the authors made two decisions: (1) to establish the *p* value at 0.20, in order to converge with the decision made upon the Egger’s Test; and (2) to use the *τ*_Kendall_ coefficient, which assesses the potential publication bias regarding the correlation size, which might be large. Both procedures were performed twice, one for cognitive empathy, and one for affective empathy. 

Meta-analytic Modeling. After that, two meta-analyses were performed, each per the Cronbach’s *α* calculated for each factor of the scale. In order to reduce the effect of the non-normality of the distribution of the Cronbach’s α coefficients, and to stabilize their variance, each Cronbach’s *α* was transformed with the Bonett method [48] due to its better theoretical correspondence [48]. Subsequently, they were transformed into the original metric of the Cronbach’s α coefficients for interpretation. Meta-analyses of Cronbach’s *α* coefficients were carried out with a random effects model, which is generally accepted and recommended for meta-analytical studies [49]. The presumption for this model is that the studies come from an overpopulation of studies that generate estimates of internal consistency. Of these studies, variability in an unknown range is possible and realistic. With this assumption, and with the estimates made with the random effects model, a greater external generalization can be achieved in cases like the one presented, in which only selected articles from a larger set are analyzed [49]. Each Cronbach’s α coefficient was weighted with the inverse of its variance. For each dimension of the BES, the meta-analytic mean of the Cronbach’s *α* coefficients was calculated using the method of estimation by restricted maximum likelihood (REML), with its confidence intervals (95% CI) adjusted by the Hartung and Knapp method [50,51].

Assessment of heterogeneity. To evaluate the heterogeneity between studies, a forest plot was performed to visually identify the dispersion of the Cronbach’s *α* coefficients, in addition to the Cochran Q statistic and the I^2^ index, which complement the Q statistic in this difference evaluation. A Q statistic with *p* < 0.05 indicates a heterogeneity beyond sampling error [52]. The practical significance of this detected heterogeneity was evaluated with the *I^2^* index, which can be around 0% if it is null, 25% if it is low, 50% if it is medium, 75% if it is high, and 100% if it is total [53]. The proposed moderators were analyzed individually to verify if they could be considered as sources of variability. A greater heterogeneity of 75% would imply a recommendation to carry out analyses to verify the effect of the moderating variables [54]. The steps applied in the study of Rubio-Aparicio et al. [37] were followed, as it was considered an updated example of good practice of a reliability generalization meta-analysis. In order to facilitate the observation of the results, descriptive statistics were calculated, taking the Cronbach’s *α* coefficient from each empathy factor from each article as the dependent variables. The independent variables considered were the positions and confidence interval values of these means, depending on whether they were below, above, or within the global mean calculated by the meta-analysis. All these calculations were done with the 1.6 version of the *jamovi* [38] software for statistical analysis.

The analysis of the moderators’ effect over reliability was based on a model of analysis of variance (ANOVA) of mixed effects (mixed-effect model, MEM), with a REML estimator. This model was chosen because the variables presented as independent variables are categorical. This analysis was performed individually for each of the moderators in each of the empathy factors, both cognitive and affective, taking into account their Cronach’s *α* coefficients as dependent variables. Three moderators were selected as possible sources of variability. According to the characteristics of the sample, the potential implications that both the type, general or special (clinical, forensic, etc.), and the generational group to which the sample belonged (adolescents, young people, or adults) were observed. Sex was not taken into consideration because only two articles presented mixed samples. Neither was the language in which the instrument was written, due to the great heterogeneity between languages, and the low representation of each one among the total of articles considered. Finally, the evaluation of potential heterogeneity was also implemented with the analysis of the studies that could influence this heterogeneity. First, these studies were identified as outliers when their values were outside the confidence interval (95%) of the meta-analytical Cronbach’s *α* [55]; second, once the outliers were identified and removed, a robust estimation of meta-analyzed Cronbach’s *α* was obtained using the same random-effects model. These last two analyses were performed with the *dmetar* [55] program. 

### 2.4. Corroboration of the Meta-Analytical Report

In order to verify that this work has been carried out according to the indications of REGEMA [29], a self-analysis was performed in which the checklist proposed by the REGEMA guide itself was completed. This checklist consists of 30 items that assess the most relevant points of the sections: title, abstract, introduction, methods, results, discussion, financing, and protocol. The answers to this table were categorical, the options being “yes” or “no”. The “not applicable” category was offered too, in case the item was not relevant for this study. In order to easily find each item’s information throughout the text, the table also included the first page in which it appeared. 

## 3. Results

### 3.1. Systematic Review

The whole search was carried out following the PRISMA method [36] for systematic reviews (Figure 1).

#### 3.1.1. Study Selection and Study Characteristics

Regarding the articles identified in the first iteration, the final number was 366. Of these, 49 were found in Scopus, 78 in WoS, 228 in PsycInfo, and 11 in Dialnet. The total number of articles found with this search (on 22 March 2021) was 366. All of them were inserted in RefWorks, a bibliography manager, which detected a total of 283 duplicates. Therefore, the abstracts of the remaining 83 articles were read. From the references included in these articles, two were highlighted as potentially appropriate. After reviewing them, both were added to the process as part of the manual search. 

The screening based on the eligibility criteria ended up with the removal of five articles. The texts of the remaining 80 articles were read. This resulted in six of them being excluded. The reasons were: not presenting the full text of the article in English or Spanish (n = 2), not contemplating the BES instrument in its study (n = 1), and being summaries of conference proceedings (n = 3). Finally, the review was carried out with a total of 74 articles that met the criteria required. 

#### 3.1.2. Results of Syntheses

Regarding the systematic review, it is observed that the use of the BES has been increasing over the years (Figure 2). Its progression is almost linear (Figure 3), with a linear correlation coefficient of *R* = 0.889 and a determination coefficient of *R*^2^ = 0.791 (*p* < 0.001; *t* = 6.45). Since it was created in 2006, 74 articles have been reported, and 68.9% of them have been published during the last five years (until 2020).

There are a great number of proposed adaptations to other languages. The most recent examples include Spain [56], Turkey [57], Iran [58], Italy [59], Peru [24], Belgium [60], and China [61], among others. According to the number of publications, two main authors are highlighted for having a high number of studies reported in which they have used the BES. Firstly, the pair of authors, formed by Malgorzata Gambin and Carla Sharp, who have four articles published in which they study a population of young mental health patients [62,63,64,65]. Secondly, Pedro Pechorro and his colleagues, who have published a total of nine articles in which the instrument BES appears [16,17,19,20,66,67,68,69,70]. This author focuses on the study of empathy in forensic samples of young Portuguese criminals.

Regarding the time sequence, there is only one study that is considered longitudinal by its authors [71], with a six-month two-time-point design. The rest of the articles have a cross-sectional design, and some of them indicate that this characteristic is one of their limitations, and they even propose the development of longitudinal studies in their future investigations [64,72,73]. Focusing on the approach, qualitative, quantitative, or mixed, the total of the articles included in this review were considered quantitative.

According to the analysis of the sample carried out with articles found since 2015 (n = 54), some patterns were observed both in the population groups and in the field of study to which each article was dedicated. Regarding gender, only 8 of the 54 articles reviewed present a sample made up of men or women exclusively (i.e., not mixed). Five of these samples are composed of women, and the other three are composed of men. With regard to generational groups, 32 studies have been conducted on the adolescent population (59.26%), 14 on adults (25.93%), 5 on young university students (9.26%), and the remaining 3 on children (5.56%). However, among the articles conducted on the adolescent population, two articles also included children, and one included adults.

When observing the field of study to which each research is directed, 22 of these 54 articles study the properties of empathy in its relation with the sample to which the BES was administered (40.74%). The next most repeated approach is the study of a clinical population, with a total of 12 articles (22.22%). Ten of them were conducted among people with mental health disorders, and one including both the patients and the healthcare providers. Seven papers are centered entirely on the study of health service workers (12.96%), with four of them focusing on the study of nurses. Ten articles study the forensic sample responses (18.52%), usually made up of young people in detention centers, and two of these ten relate their responses to the study of bullying. Two articles are devoted to the study of cyberbullying (3.7%), and one article is focused on a population of teachers (1.85%).

### 3.2. Validity Analysis

For descriptive purposes, Table 1 shows the percentages of articles that provide information related with each of the five standards proposed by the guideline “Standards” [31]. An unequal distribution throughout the articles is observed. More detailed information related to the results of each standard is presented in the following sections. This distribution was conducted by the authors of this study, according to the recommendations of the guideline, until a consensus was reached. A brief description for the presence of each validity evidence is given in Appendix A. 

#### 3.2.1. Evidence Based on Test Content

The number of articles that met the parameters for studying the first of the standards, the evidence based on test content, is 12 (57.14%). Of them, 10 reported having been translated by experts. Regarding the other two, one was reviewed by participants who had previously performed the test [25]; and the other one is the document in which the BES instrument appears for the first time, in which an explanation of how the items were created is offered [3]. On the other hand, eight articles (38.1%) did not meet these parameters. In addition, one article was considered ambiguous [74] because no explanation from the authors was given to ensure that the test they performed before administering the questionnaires was aimed at assessing the content of the instrument.

#### 3.2.2. Evidence Based on Response Processes

The second standard, the validity relative to response processes, was met by an article [75], which corresponds to 4.76% of the total. The technique used was cognitive interviews with the participants. However, two articles were considered ambiguous (9.52%). In Geng et al. [76], participants were asked for each item of the instrument to evaluate their understanding, and in Herrera-López et al. [74], the test was evaluated with 60 subjects prior to implementation with the rest of the sample. In both cases, the authors did not provide a justification to indicate that the objective was to evaluate how the participants responded to the items.

#### 3.2.3. Evidence Based on Internal Structure

The third of the standards, which studies the validity of the internal structure, was divided into five sections. The first one was the study of the factor analysis carried out in each article. The 21 articles reported having performed a confirmatory factor analysis (CFA).

The second section analyses reliability. All the studies presented reliability measures of internal consistency, mostly performed with the Cronbach’s *α* coefficient. In 4 of the 21 articles (19.05%), the analysis with the McDonald’s *ω* coefficient was also performed [17,23,24,74]. In one article’s case [73], only the analysis with the McDonald’s *ω* coefficient was performed. Four articles of the twenty-one (19.05%) carried out the test-retest technique, which corresponds to the third section of the internal structure validity tests. Of these, the study by D’Ambrosio et al. [77] performed a three-week interval between the first and second administration of the test. This is followed by Bensalah et al. [25], and Geng et al. [76], both with an interval of four weeks. Finally, in Carré et al. [26] there is an interval of seven weeks.

In the fourth section, the information pertaining to the calculation of invariance was extracted, present in 8 of the 21 articles (38.1%). Of these eight, five analyze the invariance based on groups according to the gender of the sample [73,74,78,79,80]. Of the remaining three, one analyses it based on the gender and the level of education [24], another based on the gender and age [81], and the last one based on the level of development [75]. Regarding the extent of variance studied, three studies reached the residual level [24,73,74], two the scale level [80,81], two the metric level [75,78], and one the intercepts level [79].

Finally, the fifth section of the internal structure study is responsible for checking if analyses looking to verify if there is an equivalence between different versions of the instrument were carried out. These versions could be either reductions of the original instrument, adaptations to other languages than English, or the original version that was adapted to be used in a different sample. Of the total of 21 articles, 13 (61.9%) studied the equivalence. In nine cases [3,15,17,18,25,26,66,79,82], a comparison between the original version and the reduced version that they set out within their sample was made. In three cases [75,77,83], their analysis of the instrument was compared with analyses carried out previously. In the remaining case [76], both comparisons were made within the same article.

#### 3.2.4. Evidence Based on Relations to Other Variables

The fourth standard analyzes validity according to the relationship with other variables. It is observed that the articles that reported it used convergent and discriminant evidence. Evidence is determined as convergent on the basis that the authors expect the empathy construct to correlate, positively or negatively, with another construct. Evidence is considered discriminant when it is hypothesized that empathy will not present correlation with another studied variable. For articles in which this measure was explicitly indicated by its authors, this consideration was followed. For those articles in which there was no indication, an attempt to extract the implicit information regarding the type of evidence was made. We observed one article (4.76%) that measures discriminant evidence [25], and ten (47.62%) that measure convergent evidence [15,17,18,74,75,76,78,79,80,81,82]. Seven articles (33.33%) measure both of them [3,22,58,71,72,75,84].

Social desirability appears in five of the eight articles that measure discriminant evidence with empathy, which makes it the most repeated construct [3,25,26,77,83]. Regarding the measurement of convergent evidence, the comparison with other constructs did not follow such a clear trend, ranging from comparisons with empathy [26,77,78,83], to psychopathy [17,79], bullying [3,80], social skills [18,81,82], and insensitivity [15,66], among others. 

#### 3.2.5. Evidence Based on the Consequences of Testing

Finally, the fifth standard focuses on the verification that there is evidence concerning the measurement of the consequences of having applied the test on a given sample. On this occasion, only one article [14] met this standard of validity. This article explains an implication concerning clinical practice. It indicates a conclusion once the test has been administered, that the measures applied to parents and children should be taken into consideration as complementary, never equivalent, evidence.

Appendix A presents the results of the validity tests according to the described standards, indicating, in the corresponding cases, why they are considered present.

### 3.3. Meta-Analysis

#### 3.3.1. Reliability Report

Of the 74 articles that were extracted in the review, 52 (70.27%) used the original version of the instrument, which has 20 items: 11 to measure affective empathy, and 9 for cognitive empathy. A total of 13 more versions were found, according to the total number of items and, specifically, to which items were removed from each version. With regards to the reliability report (Table 2), it is observed that 7 of the 74 articles did not report it (9.46%), 2 reported it vaguely (2.7%), and 2 precisely (2.7%). Of the remaining 63 articles (85.14%) which reported it, it was considered unusable for 12 of them (19.05%), and usable for 51 (80.95%). Finally, 34 of those 63 articles (53.97%) were classified as non-relevant.

#### 3.3.2. Reliability Levels Description

Regarding the cognitive empathy factor, 70% (n = 21) of the articles included in the meta-analysis showed values for their means of Cronbach’s *α* and their confidence intervals located above the cut-off point of >0.70, whereas only 26.66% (n = 8) were above >0.80. Just one article (3.33%) was located below the cut-off point of >0.70, and 10 (33.3%) were considered below >0.80. The rest of articles were identified as inconclusive, as the proposed cut-off points (>0.70 and >0.80) were located in between their confidence interval values. Specifically, eight articles (26.6%) were considered inconclusive when compared with >0.70, and twelve articles (40%) when compared with >0.80. Moving on now to the affective empathy factor, the 66.6% (n = 20) of the articles were considered above >0.70, 3.33% (n = 1) below >0.70, and 30% (n = 9) inconclusive. According to the next level, 43.33% (n = 13) of the articles were located above >0.80, 46.66% (n = 14) below, and 10% (n = 3) were considered inconclusive. See Appendix B.

#### 3.3.3. Reliability Generalization and General Heterogeneity Assessment

The results in relation to the potential publication biases calculated with the Egger’s Test showed that the null hypothesis was accepted on cognitive empathy, *t*(28) = −0.08, *p* = 0.93, *b* = 1.69 (95% CI: 1.07, 2.32). Regarding the Rank’s Test applied on cognitive empathy, it showed a low, although statistically significant, correlation (*τ*_Kendall_ = 0.17, *p* = 0.18). According to affective empathy, the null hypothesis was again accepted due to the results of the Egger’s Test, *t*(28) = −1.22, *p* = 0.23, *b* = 1.93 (95% CI: 1.39, 2.47). The results from the Rank’s Test did not show a statistically significant correlation (*τ*_Kendall_ = 0.04, *p* = 0.73).

The meta-analysis operations were performed twice, one for cognitive empathy, and one for affective empathy. The Cronbach’s *α* mean for cognitive empathy of the 30 articles was 0.81 (95% CI: 0.77–0.85). To evaluate the variability of Cronbach’s *α* in the different samples, heterogeneity was calculated. There is a statistically significant heterogeneity on the total sample Q (gl = 29) 2874.28, *p* < 0.0001. Using the *I*^2^ index, the variability ratio was 98.81% (>75%: high). The mean of each Cronbach’s α for all cognitive empathy studies, with their respective confidence intervals, is shown in Figure 4. Thirteen of the thirty articles present their mean values and confidence intervals below the global mean calculated by the meta-analysis, with a median of 0.68; ten articles share values with this mean, with a median of 0.795; and the values of the other seven are above that mean, with a median of 0.91.

Regarding affective empathy, the Cronbach’s *α* mean was 0.81 (95% CI: 0.76–0.84). As for the heterogeneity, a statistically significant Q value was obtained (gl = 29) 1813.65, *p* < 0.0001. Regarding the proportion of variability calculated with *I*^2^, the value obtained was 98.5% (>75%: high). The information extracted from the meta-analysis for affective empathy can be observed in Figure 5. The mean values and confidence intervals of 16 of the 30 articles are located below the global mean, with a median of 0.70; 3 articles share values with this mean, with a median of 0.83; and the remaining 11 articles have values above that mean, with a median of 0.87. Descriptive statistics for each factor are found in Table 3.

#### 3.3.4. Heterogeneity Assessment: Moderator Analysis

The ANOVA performed for the selected moderators showed that there were no statistically significant differences between the presence and absence of each moderator and the influence on the mean of Cronbach’s *α* (Table 4). For the cognitive empathy variable, the result of the analysis according to the type moderator was *F* (gl = 1.28) 1.9438, *p* = 0.1742, and according to the generation moderator, was *F* (gl = 2.27) 2.1327, *p* = 0.138. For the affective empathy variable, the result according to the type moderator was *F* (gl = 1.28) 2.7645, *p* = 0.1075, whereas for the generation moderator was *F* (gl = 2.27) 0.5476, *p* = 0.5846. Information on the 30 articles used in the meta-analysis is given in Appendix C.

#### 3.3.5. Robust Estimation

For cognitive empathy, the outlier analysis identified 16 studies (1, 4, 6, 7, 13, 14, 15, 16, 17, 18, 19, 21, 22, 24, 26, 28); for identification, see Figure 4. With the remaining 14 studies, the Cronbach’s *α* mean was 0.79 (95% CI: 0.77, 0.80), and heterogeneity was substantially reduced (see Table 5) up to be considered moderate (between 50% and 75%). Regarding the estimate of the total sample, the attenuation can be considered small (Δ_α-rob α_ 5%; see Table 5). For affective empathy, 17 outlier studies were identified (1, 2, 3, 5, 7, 8, 11, 14, 15, 16, 17, 18, 19, 20, 21, 26, 29); for identification, see Figure 5. Ten of these studies were the same as those identified in the cognitive empathy analysis. The meta-analytic alpha was 0.80 (95% CI: 0.77, 0.82), and heterogeneity barely decreased, remaining at a level which is considered high (>75%).

## 4. Discussion

The aim of this article is to carry out an in-depth analysis of the use of the BES instrument. A systematic review has been performed to synthesize the evidence of validity, and to make a reliability generalization meta-analysis. The literature search and the selection of studies were carried out according to the guidelines of the PRISMA method [36]. As can be interpreted from the systematic review, the number of articles per year has progressively increased, being most frequently used in recent years. In contrast to the most studied topics in Jolliffe and Farrington [12], which are related to crime and bullying, the trend in recent years has focused on the study of the level of empathy as a personality characteristic. This trend is followed by its application to the clinical population, which is clearly distanced from what was initially proposed by its authors.

In line with the original study, which concerns the development of the instrument [3], there is a greater focus on the adolescent population in more than half of the studies from the last five years. A good study of empathy in this population group may be of interest for the prevention of crime, due to the relationship observed between the lack of empathy in the adolescent population and conflicts with the law [12]. It should be noted that only 1 out of the 74 articles carried out a longitudinal study. However, according to the work of Delgado Rodríguez and Llorca Díaz [85], it cannot be considered longitudinal, as it does not present more than two measurements over time. This would reformulate the situation so that all studies would then be considered as cross-sectional. In contrast to cross-sectional studies, longitudinal studies demonstrate a higher statistical power [86], and due to that, not presenting longitudinal studies may imply a limitation in the sample of this study. 

Another weakness found in the sample of the articles included in the systematic review is related to the methodology used, as every single article included in the systematic review was considered quantitative. Nevertheless, the approach of mixed methods offers a higher inference quality compared with pure quantitative or qualitative methods. This inference quality is convergent with validity and data quality [87,88,89]. Mixed methods research can lead to the development of meta-inferences if the information it provides is interpreted in a holistic way [90]. Due to this, it is recommended that the production of mixed method studies is increased. 

With regards to the validity evidence analysis, it follows the method and guidelines proposed by the “Standards” guide [31]. The first work to carry out a review of the validity sources according to the standards was the study of Hawkins et al. [32]. In line with their results, most of the articles of the current review presented tests to study the third and fourth standard, leaving the rest noticeably less studied. The next most frequently reported standard was the validity of test content, although, in most cases, it is because the authors describe a translation done by experts on the subject. Some studies reported standardized methods, such as the one by Hambleton and Li [91], even though this is not one of the validity methods initially proposed by the guide. However, it was considered appropriate in this study.

The fourth standard, which is the validity based on relation to other variables, was the most reported one. This demonstrates an inclination by most authors to ensure that empathy is well defined by the BES instrument, approaching similarly considered constructs (for example, empathy measured with another instrument), and moving away from those which should be opposite (for example, sympathy). A consensus was found among most of the articles with regard to the choice of constructs, which were quite similar to what was proposed by Jolliffe and Farrington [3].

For the third standard, the subdivision into five sections facilitated the analysis of the internal structure. Although most of the sections were reported by many articles, this trend was not met for the test-retest or for the invariance calculation. Studies dedicated to the study of psychometric properties should be improved in these fields. It is observed that the total of 21 articles use reliability coefficients, such as Cronbach’s *α* or McDonald’s *ω*. This, according to the classical test theory, focuses the attention on the accuracy of the measurement. However, it has been avowed that the estimate of Cronbach’s *α* should be replaced by McDonald’s *ω*, because *ω* represents a more realistic model of how the relationship between items and its construct is expressed (i.e., congeneric) [92,93].

The evidence of measurement invariance is a sine qua non requirement for group comparison and for reducing the bias in the analysis of group differentiation, caused by the different structural properties of the BES. Since this property is one that is infrequently performed in BES studies, and the comparison between groups is routine, it is likely that some of these differences include irrelevant variability due to the different psychometric properties. This argument leads to a practical implication: the dimensionality corroboration and the group equivalence must be assessed in substantive studies, due to the fact that these are not static properties, and they are directly related to the reliability estimation [92]. This corroboration can be considered even more strict when the instrument has been derived from a process of adaptation from another cultural context [91].

On the other hand, the evaluation of dimensionality was usually carried out by a confirmatory approach (i.e., CFA). However, in recent years, the apparent standard for properly assessing dimensionality has been exploratory structural equation modelling (ESEM) [94], which has been conceptually and empirically a highly recommended procedure for assessing the internal structure of psychological measures. This is because it overcomes the limitations of the CFA in estimating its parameters [94,95].

Two main conclusions result from the descriptive analysis based on the “Standards”. First, dimensionality assessment was one of the most reported validity indicators. Its analysis showed two robust dimensions. This result strengthens the underlying theory, and enables intercultural comparisons. Second, evidence based on relation to other variables were consistent with prior expectations and post-tests assessments. This gives support to the theoretical framework on which the BES is based. It indicates that the BES has a high power to measure empathy and empathy’s relationship with other psychosocial factors. 

Considering standards to measure validity are especially interesting nowadays, the World Health Organization (WHO) calls for the development and application of standardized science-based methods to ensure good practice in the field of health [96]. Due to the increased standards required for the publication of scientific literature in health, there are guidelines, such as the Minimum Information for Biological and Biomedical Investigations [97] and Enhancing the Quality and Transparency of Health Research [98], which provide a framework to promote the transparency of the methods used. Another guideline is proposed by the Journal Article Reporting Standards (JARS) group, which is part of APA. Within the recommendations on psychometry, it is considered essential to introduce reliability and validity tests [99], something that has not always been observed in the studies used in this meta-analysis.

The results extracted from the assessment of the potential publication biases showed that the null hypotheses were not rejected in both the parametric and non-parametric tests used. Additionally, the effect size of this potential bias measured by the non-parametric test of Rank (using *τ*_Kendall_) was close to zero. This does not mean that there is an absolute absence of biases. In fact, this information only regards the results of the statistical tests applied (Egger’s and Rank’s). However, the bias size that was found can be interpreted as just a sampling error.

In addition, it is important to keep in mind that the studies included in this systematic analysis which reported their reliability coefficients were not limited to those in which scores were located above 0.70 or 0.80 cut-offs. Also, only those articles which reported their Cronbach’s *α* means for cognitive and affective empathy were included in this meta-analysis, so the reliability induction (reporting prior studies’ data [33]) was reduced. Due to that, the measurement error size can be truthfully interpreted in order to analyze the reliability scores.

The means of Cronbach’s α obtained in the meta-analysis are considered satisfactory for the comparison between groups, because both cognitive and affective empathy means are between 0.70 and 0.80. These results may not be appropriate for clinical use, in which case measures focus on particular individuals, and, therefore, require measures of at least 0.90, an average of 0.95 being considered desirable [100]. High levels of reliability indicate lower magnitude of measurement error, which is more desirable in clinical contexts.

On the other hand, there is a great heterogeneity between the means and confidence intervals of the articles of the meta-analysis, results that are repeated in last year’s studies that carried out reliability generalization meta-analyses in the field of psychology [101,102,103]. In the present study, about half of the articles presented mean values and confidence intervals of Cronbach’s *α* located below the global mean, both for cognitive empathy, which are 43.3% of the total, and affective empathy, 53.3% of the total. Only about one third of the articles are above this mean, i.e., there is a low number of articles with fairly high mean values, which indicates that most of the articles that have reported the reliability of the BES with 20 items have low levels of reliability.

According to Molina Arias [104], a great heterogeneity implies that a good analysis of the moderators must be carried out. However, concerning the moderators, there is no significant difference between the sample considered general and the one considered special, which is composed of a clinical or forensic population. There are also no differences between the studies carried out in the adolescent, young, or adult population. This indicates that heterogeneity cannot be explained by such moderators, so it may be necessary to specify the type of sample, distinguish between clinical or forensic populations, or even analyze other moderators that could be interesting, such as the language of the instrument.

Regarding the corroboration of the meta-analytical report, it was considered very supportive to have a checklist when following the steps set out in the REGEMA guide [35]. This study presents not only good replicability, but also good reproducibility as promoted by López-Ibáñez and Sánchez-Meca [105], which makes it possible for any other researcher to repeat the calculations made in it, including the same data.

### 4.1. Limitations and Future Research

With respect to the limitations of this study, the great heterogeneity within the forms in which the BES is presented is highlighted. The language, the number of items, and the factors in which they are divided, taking also into consideration the adaptations of certain countries that present their reduced and translated version, are some of the examples in which the heterogeneity is observed. That, together with the fact that not all articles have measured their reliability, or that they have measured it by using different coefficients (McDonald’s *ω* or Cronbach’s *α*), has led to the total number of studies to be included in the meta-analysis to be less than half of the total number of studies found in the first systematic review. Therefore, it has not been possible to carry out a screening based on characteristics that homogenize such a total number of articles, such as the quality of the statistical analyses, the number of subjects that form the sample, the language of drafting of the instrument, the sex of the sample, the time sequence of the study (longitudinal or cross sectional), etc. 

On the other hand, the lack of analysis of the articles in terms of possible biases that may affect their reliability levels, such as response patterns, type or sample size, etc., make it difficult to interpret the great heterogeneity found in this study. In particular, some of the required aspects advised to be assessed in each study are response patterns, such as insufficient effort or neglected answers, expressed as excessive consistency or inconsistency, as well as the appropriate model to measure the answers (congeneric or tau-equivalent [92,93]). Another limitation is the inter-evaluator reliability procedure. Said approach was not conducted, and the possible disparities between researchers were managed following a qualitative procedure. Other quantitative procedures, such as the Kappa coefficient [106], are much more recommended.

Regarding the theoretical framework explained in this study, there is a lack of citation of some authors that have made relevant contributions to the emotional component of empathy. An example is the case of Antonio Damasio and his group, who have recently been focused on the relationship between empathy, emotions, and feelings when listening to sad music [107,108,109,110,111]. Another example is Rosalind Picard and her team, who have been studying emotions through a perspective based on the combination of psychology and engineering with the use of devices, artificial intelligence, mobile sensors, etc. [112,113,114,115]. Finally, authors such as Andrew Ortony [116] and Ira Roseman [117], and their respective colleagues, have dwelled in the study of emotions. It is pointed out that an in-depth study of these authors will help to improve the quality of research based on empathy and emotions. Also, the inclusion of their emotional models in future research is considered interesting.

The last limitation is based on the meta-analysis itself. It is only focused on the reliability construct. On one hand, opening the range of constructs in which to conduct a meta-analysis would be interesting. For example, performing a validity meta-analysis could have enhanced these results. On the other hand, the results of the present study cannot provide highly practical implications due to the lack of statistical significance provided by the moderators’ analysis. That is why, for future research, it would be of interest to carry out an extension and review of moderators that can act as variables which hinder the reliability generalization of the instrument. 

More studies analyzing the properties of the BES scale would help to homogenize the results. Therefore, repeating this meta-analysis in the future may be a good indication. Finally, the robust estimates of the Cronbach’s *α* coefficient in both subscales, once more than 10 studies identified as outliers were removed, were not significantly different from the estimate in the total sample of studies. This indicates that heterogeneity was still present, although in smaller magnitude in the cognitive empathy subscale. In contrast, in the affective empathy subscale, heterogeneity barely decreased when outliers were removed, indicating that the Cronbach’s *α* mean can still be considered not robust. A characteristic pattern of these studies identified as outliers is not clearly observed, but since the extreme values generally do not ensure an accurate estimation of any statistical parameter, there is still a gap to find out which are the causal factors that provoke the variability of reliability in the BES. 

### 4.2. Practical Implication

The relationship between empathy and antisocial behavior has been of great interest in psychological research over time [12,13]. A good study of the instruments that measure constructs such as empathy can be of great help both in the field of prevention and psychological intervention in social areas. This analysis of the BES instrument aims to contribute in facilitating working with the general population, in such a way that prosocial behaviors are increased while disruptive ones are reduced. Therefore, a professional psychologist can take advantage of this study if a positive result in the assessment of the BES is obtained. With this information, the professional can, among other proposals, perform empirical studies in which the empathy variable is correlated with other variables of interest, measure the basal level of empathy among a sample in which a social intervention can be done, or assess the effects that an intervention has had on the empathy of a population. In addition, the results obtained can be generalized when a good degree of confidence is obtained. That is because, in order to generalize the results to different future studies, the random coefficients model is generally accepted as the recommended option. This is one of the preferred research goals [118].

This study is not limited to assessing the suitability of the BES for its implementation, but also seeks to increase the value of the validity and reliability standards recommended for the research of instruments focused on health. Deepening the standard of internal structure validity, together with the study of the meta-analysis of reliability, has allowed delving into the determinants that ensure equivalence between sample groups. On the other hand, an attempt has been made to question the widespread use of the Cronbach’s *α* coefficient as opposed to the McDonald’s *ω* coefficient, encouraging the authors to model the data in order to ensure the appropriate choice of one or the other [92]. Not only in the choice of coefficients, but also in the evaluation of dimensionality, it is proposed to make an assessment of the suitability for the use of the ESEM model versus the AFC, or the use of both. This is because, apparently, the former is more recommended in psychological measures [94,95]. 

It is intended to appeal to the authors of scientific articles in which instruments of any variable are measured, by encouraging the reporting of reliability, even in non-psychometric studies. This will have practical implications both for other authors and for reviewers, as it is intended to establish the reliability standard. In other words, this article aims to promote transparency both in the methods and results carried out in the use of an instrument or scale in the field of health, as recommended in guides [97,98] and organizations [96,99] that promote good practice.

## 5. Conclusions

This paper takes a novel approach to the analysis of the BES instrument throughout its implementation. Both the validity tests and the meta-analysis show that much of the sample of studies drawn from the systematic review present a lack of data that would facilitate a good interpretation of the generalization of their reliability. However, based on the results obtained, it is noted that the BES instrument in its original version presents good values to be used in the measure of empathy in general population groups. Future research would be needed to assess whether its use in clinical diagnosis is also correct, which has been dismissed by the authors according to the results of this study. This is considered a highly relevant study due to the fact that lack of empathy has been historically related to crime, as was clearly detailed in the work of Jolliffe and Farrington [14]. Therefore, being able to ensure that a scale that assesses empathy is valid and reliable is considered of great value for society by the authors of this study. Also, the concept of empathy is, as has been shown, constantly being defined [2], a task this article can help in doing.

## Figures and Tables

**Figure 1 healthcare-10-00029-f001:**
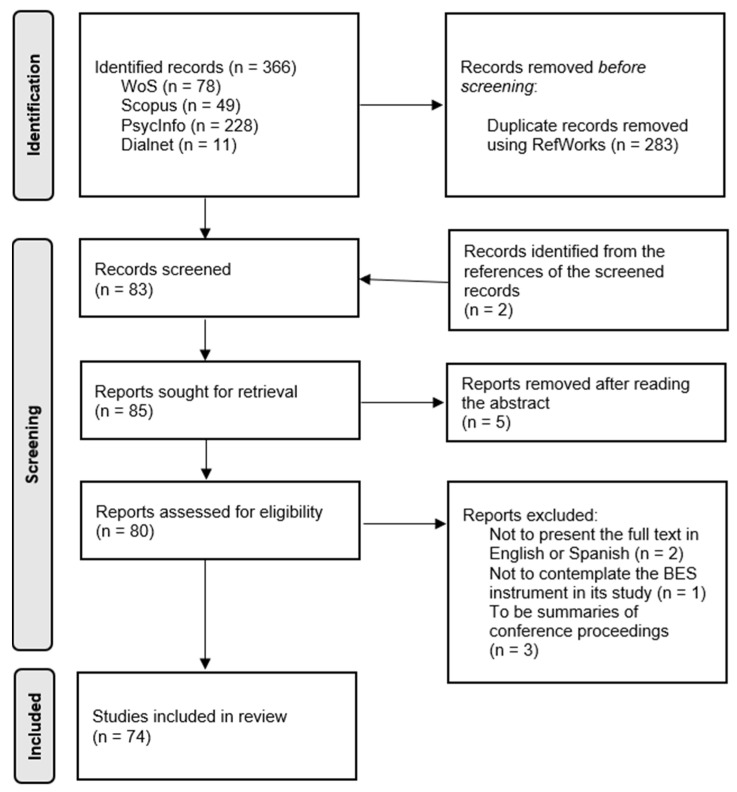
PRISMA flow diagram of article selection process.

**Figure 2 healthcare-10-00029-f002:**
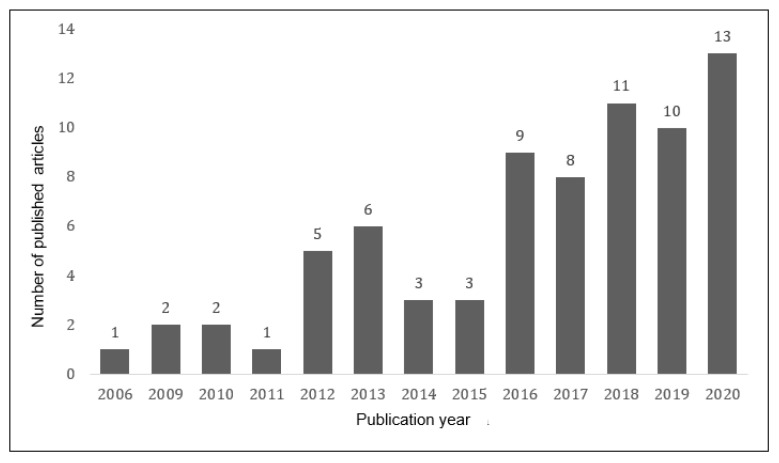
Number of publications that include the BES instrument over time.

**Figure 3 healthcare-10-00029-f003:**
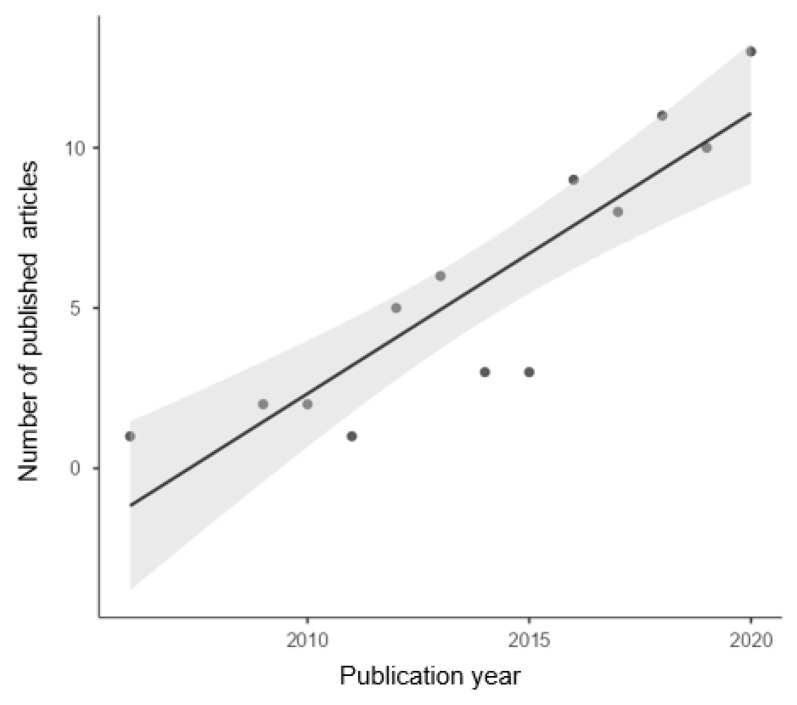
Scatterplot of articles published over time.

**Figure 4 healthcare-10-00029-f004:**
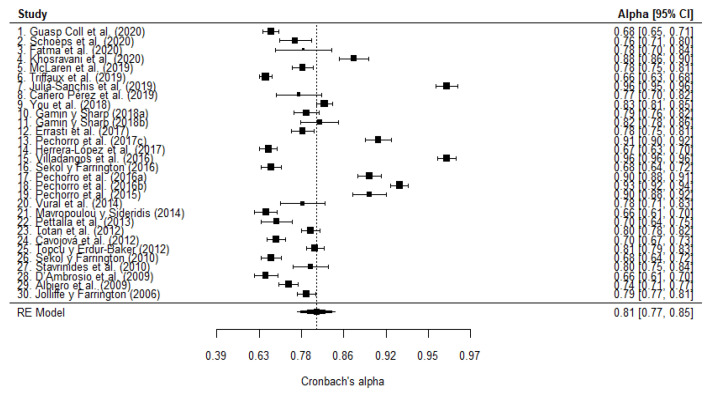
Forest plot of the cognitive empathy meta-analysis.

**Figure 5 healthcare-10-00029-f005:**
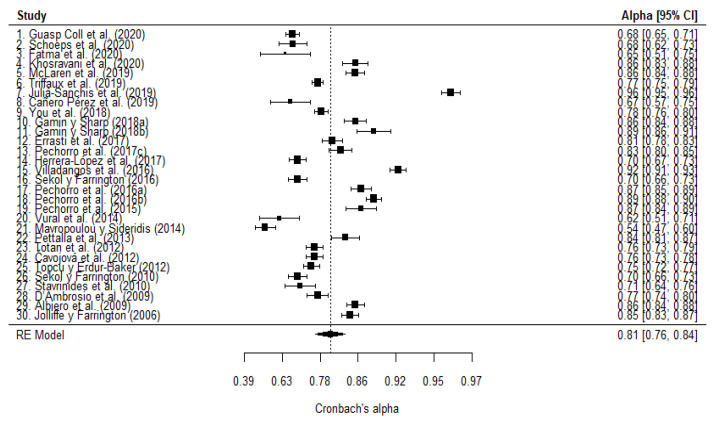
Forest plot of the affective empathy meta-analysis.

**Table 1 healthcare-10-00029-t001:** Number of studies and percentages for each validity test.

Study	Test Content	Response Processes	Internal Structure					Relation to Other Variables	Consequences of Testing
			Factor Analysis	Reliability	Test-Retest	Invariance	Equivalence between Versions		
Yes	12 (57.14%)	1 (4.76%)	21 (100%)	21 (100%)	4 (19.05%)	8 (38.1%)	13 (61.9%)	18 (85.71%)	1 (4.76%)
No	8 (38.1%)	18 (85.71%)	0	0	17 (80.95%)	13 (61.9%)	8 (38.1%)	3 (14.29%)	20 (95.24%)
Ambiguous	1 (4.76%)	2 (9.52%)	0	0	0	0	0	0	0

**Table 2 healthcare-10-00029-t002:** Reliability report.

Number of Items (BES Version)	Induced Reliability			Reported Reliability		
	Omitted	Vague	Precise	Unusable	Usable	NR
20 (original)	7	2	2	9	32	26
20 (adaptation to “victim”)	-	-	-	-	1	-
40	-	-	-	1	-	1
19 (item 4 is removed)	-	-	-	-	1	-
18 (items 1 and 6 are removed)	-	-	-	-	1	-
18 (items 4 and 15 are removed)	-	-	-	-	1	1
16 (items 2, 3, 4, and 15 are removed)	-	-	-	-	1	1
16 (items 4, 5, 15, and 19 are removed)	-	-	-	-	1	-
12 (items 1, 4, 6, 7, 13, 15, 19, and 20 are removed)	-	-	-	-	1	-
12 (Polish version)	-	-	-	-	1	1
9 (Spanish version)	-	-	-	1	9	2
7 (El Salvador version)	-	-	-	1	-	1
18 (items 4 and 7 are removed); 17 (adaptation to parents, in third person; items 4, 6, and 7 are removed)	-	-	-	-	1	-
20 and 7 (original and reduced version)	-	-	-	-	1	1
TOTAL	7	2	2	12	51	34

*NR* = Non-relevant.

**Table 3 healthcare-10-00029-t003:** Descriptive statistics of Cronbach’s *α* for each article according to its position with respect to the global mean.

Position Relative to the Global Mean		Number of Articles (% of the Total)	Mean	Median	Standard Deviation	Minimum	Maximum
Cognitive empathy	Below	13 (43.3)	0.704	0.680	0.0454	0.660	0.780
	Shared	10 (33.3)	0.797	0.795	0.0189	0.770	0.830
	Above	7 (23.3)	0.920	0.910	0.0311	0.880	0.960
Affective empathy	Below	16 (53.3)	0.703	0.700	0.0643	0.540	0.780
	Shared	3 (10)	0.827	0.830	0.0153	0.810	0.840
	Above	11 (36.7)	0.881	0.870	0.0330	0.850	0.960

**Table 4 healthcare-10-00029-t004:** Results of the ANOVA for moderators.

Variable	*F* (*df*)	*Q_E_* (*df*)	*R* ^2^	*I* ^2^
Cognitive empathy				
Type	1.94 (1.28)	2742.25 * (28)	3.07%	98.77%
Generation	2.13 (2.27)	2436.63 * (27)	7.57%	98.7%
Affective empathy				
Type	2.76 (1.28)	1742.55 * (28)	5.51%	98.4%
Generation	0.55 (2.27)	1452.61 * (27)	0.0%	98.51%

*df* = degrees of freedom; *F* = statistic to measure significance, according to Knapp–Hartung; *Q**_E_* = statistic to measure specification error; *R*^2^ = proportion of variance accounted by the predictor; *I*^2^ = proportion of heterogeneity. * *p* < 0.001.

**Table 5 healthcare-10-00029-t005:** Analysis and robust estimation of the meta-analytical Cronbach’s *α* coefficient.

	Cognitive Empathy	Affective Empathy
Sample		
N	14	13
N_remov_	16	17
Robust estimation		
M_rob_	0.79	0.80
se	0.03	0.06
95% CI	(0.77, 0.80)	(0.77, 0.82)
Z	45.22 *	24.33 *
Δ_α–rob α_	−0.2 (−2.4%)	−0.1 (−1.2%)
Heterogeneity		
Q (df)	41.82 * (13)	148.13 * (12)
tau^2^	0.009	0.05
Tau	0.093	0.22
I^2^	65.47%	93.47%
H^2^	2.9	12.32

*N*: number of studies included in the analysis; *N_remov_*: number of studies removed; *M_rob_*: robust mean alpha (without outlier studies), *se*: standard error; *Z*: z test. Δ*_α–rob α_*: difference between *M_α_* in the full sample and *M_rob α_*. * *p* < 0.01.

## Data Availability

Analysis script is available on request from the authors.

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
