# Peer review of "Basic Empathy Scale: A Systematic Review and Reliability Generalization Meta-Analysis"

_healthcare, 2021, doi:10.3390/healthcare10010029_

Round 1

Reviewer 1 Report

A number of quotes from well-known authors in modeling or expressing empathy and emotions were not mentioned at all in this study while these authors are referenced in PsycInfo

For example:
Damasio Antonio 205 quotes
Ortony Andrew 56 quotes
Collins Allan 164 quotes
Close Gerald 119 quotes
Picard Rosalind 37 quotes
Roseman Ira J. 27 quotes

These authors who proposed well-known emotional models were omitted by the authors as none of them are cited in the article.

Author Response

REVIEWER 1

Point 1: A number of quotes from well-known authors in modeling or expressing empathy and emotions were not mentioned at all in this study while these authors are referenced in PsycInfo

For example:
Damasio Antonio 205 quotes
Ortony Andrew 56 quotes
Collins Allan 164 quotes
Close Gerald 119 quotes
Picard Rosalind 37 quotes
Roseman Ira J. 27 quotes

These authors who proposed well-known emotional models were omitted by the authors as none of them are cited in the article.

Response 1: First of all, thank you for your kind words. We would like to also thank you for your job in this study.

Regarding the authors you proposed, we would like to cite some of their works in our manuscript, as we also believe that not taking them into account could be considered as a limitation. The new text that includes their works can be found in “4.1 Limitations and future research”, as follows:

Reviewer 2 Report

The paper with the theme “Basic Empathy Scale: Meta-Analysis of Reliability Generalization of the Instrument” is relevant and current. In general, the paper is well prepared and presents a scientific coherence between the topic and the conclusions. A detailed description of the methodology and statistical methods used is one of the strengths of this article.

However, the beginning of the “Introduction” section should begin by describing the importance of this study. At the end of the introduction, authors are suggested to formulate a starting question that is common to all papers that were subject to literature review. At the end of the "Introduction" section, the authors formulated a hypothesis "the hypothesis is that the BES instrument has good values of validity and internal consistency, so that it becomes an appropriate scale for the measurement of empathy in the general population" which I suggest that should be reviewed in light of the results achieved.

Given the quality of the results obtained and their discussion, I believe that the conclusions or highlights can be improved.

Author Response

REVISOR 2

Revisor: El artículo con el tema “Escala de empatía básica: metaanálisis de la generalización de confiabilidad del instrumento” es relevante y actual. En general, el artículo está bien elaborado y presenta una coherencia científica entre el tema y las conclusiones. Una descripción detallada de la metodología y los métodos estadísticos utilizados es uno de los puntos fuertes de este artículo.

Autores: Gracias por sus amables palabras. Agradecemos tu comentario.

Punto 1: Sin embargo, el comienzo de la sección "Introducción" debe comenzar describiendo la importancia de este estudio. Al final de la introducción, se sugiere a los autores que formulen una pregunta inicial que sea común a todos los artículos que fueron objeto de revisión de la literatura. Al final del apartado "Introducción", los autores formularon una hipótesis "la hipótesis es que el instrumento BES tiene buenos valores de validez y consistencia interna, por lo que se convierte en una escala adecuada para la medición de la empatía en la población general" que Sugiero que se revise a la luz de los resultados obtenidos.

Respuesta 1: Agradecemos sus recomendaciones. Los hemos incluido en la sección "Introducción". Esperamos que quede más claro que antes. El nuevo texto se puede leer de la siguiente manera:

“El presente estudio se considera de gran relevancia ya que puede proporcionar un análisis en profundidad de la Escala de Empatía Básica (BES) [3], así como un estudio cualitativo y cuantitativo de su validez y confiabilidad. Las revisiones y los metanálisis no se encuentran comúnmente en la literatura científica, sin embargo, los autores de este estudio los consideran importantes para ayudar a los investigadores y profesionales ”.

Esta Pregunta de Investigación (RQ) y esta Hipótesis (H) se formulan de la siguiente manera:

RQ: ¿El BES proporciona suficientes niveles de validez y confiabilidad para asegurar su uso apropiado?

H: Debido a la evidencia del uso generalizado del BES, planteamos la hipótesis de que presentaría buenos niveles de validez y consistencia interna para su uso en la población general ”.

Punto 2: Dada la calidad de los resultados obtenidos y su discusión, creo que se pueden mejorar las conclusiones o destaques.

Respuesta 2: reconocemos su recomendación. Reorganizamos parte del texto gracias a sus comentarios, por lo que incluimos este párrafo al final de la sección “Conclusiones”. Creemos que señalará la importancia que consideramos este estudio, además de mejorar las conclusiones generales. Este párrafo es:

“Este se considera un estudio de gran relevancia debido a que históricamente la falta de empatía se ha relacionado con la delincuencia, como se detalla claramente en el trabajo de Jolliffe y Farrington [14]. Por tanto, poder asegurar que una escala que evalúa la empatía es válida y fiable, es considerado de gran valor para la sociedad por los autores de este estudio. Además, el concepto de empatía está, como se ha demostrado, en constante definición [2], por lo que este artículo puede ayudar en esta tarea ”.

Reviewer 3 Report

Thank you for giving me the chance to review the manuscript "healthcare-1491036" titled "Basic Empathy Scale: A Reliability Generalization Meta-Analysis". The authors conducted a systematic review of the psychometric properties of the Basic Empathy Scale and a meta-analysis of its reliability. These type of studies are well needed in the psychometrics literature, being only seldomly conducted. However, there are some concerns upon which the authors might want to reflect on.

First of all, I have had a very hard time reading the manuscript and it was highly effortful to get a grasp of the actual work conducted by the authors. It seems like the paper was first written in the authors’ native language and then translated into English but without taking into consideration the structural differences between the two languages. Unfortunately, in the current form the manuscript overshadows the hard work put forward by the authors. I am not a native speaker myself and know from my own experience how tricky the subtleties of language can be. But I think it will benefit your paper if you get editing help from someone with full professional proficiency in English. I am afraid that in the current form the manuscript cannot be published (there are entire paragraphs I do not think I entirely or correctly understood).

Another major point regards the actual aim of the review. It seems that your work represents a hybrid approach, by conducting the systematic review on all aspects of the scale’s psychometric properties, but the meta-analysis was done only on reliability (i.e., internal consistency). Would have been a more insightful to offer a quantitative overview (meta-analyses) also for the validity of the scale. Or, since the meta-analysis on the reliability estimates did not actually bring much novelty (none of the moderators were statistically significant), another suggestion would be just to stick with the systematic review (without the meta-analytic component).

Also, regarding the aim of the study, the title is not reflecting the full aim (systematic review of psychometric properties and meta-analysis on reliability), and neither the abstract (were you almost exclusively focus on the meta-analytical findings).

When you present the inclusion and exclusion criteria you mention, among other, including: "c) experimental, quasi-experimental, or prevalence studies," and "d) validations, adaptations, and structural analyses of the BES in any language". Thus, it is not quite clear on what type of publications you focused your search. If these were experimental, then those studies focused on testing causal hypotheses by manipulating (IV) or measuring (DV) empathy, but without a focus on its psychometric properties (besides, hopefully, reporting the scale’s reliability). If you considered prevalence studies, then again, it is hard to imagine that these had also the goal of inspecting the psychometric properties of the instrument. On the other hand, if you focused on the criterion you mention at "d)" then most probably these were descriptive-analytic (correlation) with methodological focus (scale adaptations, further validations, etc.). Hence, please clearly state what types of studies you confused the selection process on, and what you actually included in your review.

You mention applying the PRISMA standards, but this is not entirely correct. For example, in the "Method" section you should explain how you conducted the search process, by using the following subsections: "Eligibility criteria", "Information sources", "Search strategy", and "Selection process"; while in the "Results" section you report the actual results of the search process (e.g., the number of retrieved articles, how many were eligible, etc.), through a first subsection with the title "Study selection”. Also, here you have to introduce the PRISMA flow chart (I also recommend you download their actual template and only adapt it for your review). Thus, please download the PRISMA 2020 Checklist (http://prisma-statement.org/documents/PRISMA_2020_checklist.pdf) and try to comply to it as closely as possible.

When you talk about the inter-evaluator reliability, you just mention how the disparities were settled. Could you please provide quantifications for each criteria (e.g., Kappa coefficients)?

A critical overview of BES’s psychometric properties is also needed in the "Discussion" section. To certain extant the authors covered it, but rather in terms of "what is lacking or still needed" and with a smaller emphasis in terms of "what are the most robust proofs in support of the scales validity".

Author Response

REVIEWER 3

Reviewer: Thank you for giving me the chance to review the manuscript "healthcare-1491036" titled "Basic Empathy Scale: A Reliability Generalization Meta-Analysis". The authors conducted a systematic review of the psychometric properties of the Basic Empathy Scale and a meta-analysis of its reliability. These type of studies are well needed in the psychometrics literature, being only seldomly conducted. However, there are some concerns upon which the authors might want to reflect on.

Authors: We would like to acknowledge your work as a reviewer. Thank you for your nice words and your recommendations.

Point 1: First of all, I have had a very hard time reading the manuscript and it was highly effortful to get a grasp of the actual work conducted by the authors. It seems like the paper was first written in the authors’ native language and then translated into English but without taking into consideration the structural differences between the two languages. Unfortunately, in the current form the manuscript overshadows the hard work put forward by the authors. I am not a native speaker myself and know from my own experience how tricky the subtleties of language can be. But I think it will benefit your paper if you get editing help from someone with full professional proficiency in English. I am afraid that in the current form the manuscript cannot be published (there are entire paragraphs I do not think I entirely or correctly understood).

Response 1: The document has been reviewed by a native English speaker to improve the quality of the text.

Point 2: Another major point regards the actual aim of the review. It seems that your work represents a hybrid approach, by conducting the systematic review on all aspects of the scale’s psychometric properties, but the meta-analysis was done only on reliability (i.e., internal consistency). Would have been a more insightful to offer a quantitative overview (meta-analyses) also for the validity of the scale. Or, since the meta-analysis on the reliability estimates did not actually bring much novelty (none of the moderators were statistically significant), another suggestion would be just to stick with the systematic review (without the meta-analytic component).

Response 2: We completely agree with your comment. It will definitely help us to improve our methodology in future meta-analyses. Thank you. In this specific case, we have decided to include it as a limitation and as proposal for future research. We have added this text in the “4.1 Limitations and future research” section:

“The last limitation is based on the meta-analysis itself. It is only focused on the reliability construct. On one side, opening the range of constructs in which to conduct a meta-analysis would be interesting. For example, to do a validity meta-analysis could have enhanced these results. On the other side, the results of the present study can not provide a highly practical implication due to the lack of statistical significance provided by the moderators’ analysis. That is why, for future research, it would be interesting to carry out an extension and review of moderators that can act as variables which hinder the reliability generalization of the instrument.”

Point 3: Also, regarding the aim of the study, the title is not reflecting the full aim (systematic review of psychometric properties and meta-analysis on reliability), and neither the abstract (were you almost exclusively focus on the meta-analytical findings).

Response 3: Thank you for your recommendation, we have changed the title of the manuscript and its abstract:

“Basic Empathy Scale: A Systematic Review and Reliability Generalization Meta-Analysis”

“Abstract: The Basic Empathy Scale (BES) has been internationally used to measure empathy. A systematic review including 74 articles that implement the instrument since its development in 2006 was carried out. Moreover, an evidence validity analysis and a reliability generalization meta-analysis were performed to know if the scale presented the appropriate values to justify its application. Results from the systematic review showed that the use of the BES is increasing, although the research areas in which it is focused are currently being broadened. The validity analyses indicated that both the type of facor analysis and reliability are much more reported in validation studies than consequences of testing is. Regarding the meta-analysis results, the mean of α for cognitive empathy was 0.81 (95% CI: 0.77-0.85), with high levels of heterogeneity (I2 = 98.81%). Regarding affective empathy, the mean of α was 0.81 (95% CI: 0.76-0.84), with high levels of heterogeneity. It was concluded that BES is appropriate to be used in general population groups, although not recommended for clinical diagnosis; and there is a moderate to high heterogeneity in the mean of α. The practical implications of the results in mean estimation and heterogeneity are discussed.”

Point 4: When you present the inclusion and exclusion criteria you mention, among other, including: "c) experimental, quasi-experimental, or prevalence studies," and "d) validations, adaptations, and structural analyses of the BES in any language". Thus, it is not quite clear on what type of publications you focused your search. If these were experimental, then those studies focused on testing causal hypotheses by manipulating (IV) or measuring (DV) empathy, but without a focus on its psychometric properties (besides, hopefully, reporting the scale’s reliability). If you considered prevalence studies, then again, it is hard to imagine that these had also the goal of inspecting the psychometric properties of the instrument. On the other hand, if you focused on the criterion you mention at "d)" then most probably these were descriptive-analytic (correlation) with methodological focus (scale adaptations, further validations, etc.). Hence, please clearly state what types of studies you confused the selection process on, and what you actually included in your review.

Response 4: We thank you for pointing this out. Actually, you are right, and we made a mistake in this description. We did not include prevalence studies, so we are going to eliminate it from the list. Regarding “d)”, we agree that it is better explained with your words, so we would like to improve the description of the item. Also, we included this text to explain the main reason:

“Furthermore, the inclusion of both experimental and correlational studies helped the researchers to have a wide overview of the BES. Although it seems that this could act as a distractor from the main focus of this review, it provided the necessary reports to conduct the further analysis of validity and the reliability meta-analysis.”

Point 5: You mention applying the PRISMA standards, but this is not entirely correct. For example, in the "Method" section you should explain how you conducted the search process, by using the following subsections: "Eligibility criteria", "Information sources", "Search strategy", and "Selection process"; while in the "Results" section you report the actual results of the search process (e.g., the number of retrieved articles, how many were eligible, etc.), through a first subsection with the title "Study selection”. Also, here you have to introduce the PRISMA flow chart (I also recommend you download their actual template and only adapt it for your review). Thus, please download the PRISMA 2020 Checklist (http://prisma-statement.org/documents/PRISMA_2020_checklist.pdf) and try to comply to it as closely as possible.

Response 5: Thanks for your comment, we have upgraded everything related with the systematic review and we followed PRISMA’s latest version, which is:

Page, M. J., McKenzie, J. E., Bossuyt, P. M., Boutron, I., Hoffmann, T. C., Mulrow, C. D., ... Moher, D. (2021). The PRISMA 2020 statement: an updated guideline for reporting systematic reviews. Bmj, 372. https://doi.org/10.1136/bmj.n71

Point 6: When you talk about the inter-evaluator reliability, you just mention how the disparities were settled. Could you please provide quantifications for each criteria (e.g., Kappa coefficients)?

Response 6: We agree that this is a limitation of our results. We would like to include a more detailed explanation of our procedure in the “Methods” section, as well as, describe it as a limitation in its section. The new texts can be read as follows:

“Disparities between researchers were estimated following a qualitative procedure. As a result, they were considered minor and easy to overcome.” 

“Another limitation is the inter-evaluator reliability procedure. Said approach was not conducted and the possible disparities between researchers were managed following a qualitative procedure. Other quantitative procedures, such as the Kappa coefficient, are much more recommended.”

Point 7: A critical overview of BES’s psychometric properties is also needed in the "Discussion" section. To certain extant the authors covered it, but rather in terms of "what is lacking or still needed" and with a smaller emphasis in terms of "what are the most robust proofs in support of the scales validity".

Response 7: We have included a new paragraph to facilitate the interpretation of the results in the “Discussion” section. Thanks for your recommendation.

“Two main conclusions result from the descriptive analysis based on the “Standards”. First, dimensionality assessment was one of the most reported validity indicators. Its analysis showed two robust dimensions. This result strengthens the underlying theory and enables intercultural comparisons. Second, evidence based on relation to other variables were consistent with prior expectations and post-tests assessments. This gives support to the theoretical framework on which the BES is based. It indicates that the BES has a high power to measure empathy and empathy’s relationship with other psychosocial factors.”

Round 2

Reviewer 3 Report

The article improved considerably. Therefore, I recommend its publication.

This manuscript is a resubmission of an earlier submission. The following is a list of the peer review reports and author responses from that submission.

Round 1

Reviewer 1 Report

Dear colleagues, I hope this message find you well.

Thank you for giving me the opportunity of reading the work “Basic Empathy Scale: Meta-Analysis of Reliability Generalization of the Instrument”, it has been a very big pleasure to collaborate reviewing this manuscript. The paper is interesting and well written. The topic of this research is very interesting and I strongly recommend to publish it. There are only four suggestions:

  1. The introduction is quite appropriate, congratulations. However, I recommended to divide the introduction into several subsections in order to do it more intuitive.
  2. Considering the specificities of the journal - which has a multidisciplinary readership - I strongly recommend the authors to introduce better why empathy could be a high relevant variable in the general population as well as the theoretical basis of the construct.
  3. “Four future research” Please, check spelling errors.
  4. In my humble opinion, it could be useful to describe in more detail the practical implications of this research. For instance, why this publication is useful and how could help psychologist?

Author Response

Response to Reviewer 1 Comments

Reviewer 1: Thank you for giving me the opportunity of reading the work “Basic Empathy Scale: Meta-Analysis of Reliability Generalization of the Instrument”, it has been a very big pleasure to collaborate reviewing this manuscript. The paper is interesting and well written. The topic of this research is very interesting and I strongly recommend to publish it.

Authors: We would like to thank you for your job and your kind words.

Point 1: The introduction is quite appropriate, congratulations. However, I recommended to divide the introduction into several subsections in order to do it more intuitive.

Response 1: Thank you for your appreciation of the introduction. We came to the agreement of dividing it in different subsections, as you proposed it. The four that were chosen were:

1.1. The development of the BES.

1.2. Description of the scale.

1.3. Proposals for the study of the scale properties.

1.4. The current study.

Point 2: Considering the specificities of the journal - which has a multidisciplinary readership - I strongly recommend the authors to introduce better why empathy could be a high relevant variable in the general population as well as the theoretical basis of the construct.

Response 2: We appraise your comment, we will strengthen the argument in order to highlight the relevance of the topic. This is new text added to this purpose:

“This is considered a highly relevant study due to the fact that the lack of empathy has been historically related with crime, as it was clearly detailed in the work of Jolliffe and Farrington [14]. Therefore, being able to ensure that a scale that assesses empathy is valid and reliable, is considered of great value for society by the authors of this study. Also, the concept of empathy is, as it has been shown, constantly being defined [2], so this article can help in this task.”

Point 3: “Four future research” Please, check spelling errors.

Response 3: This spelling error is now corrected and the whole document is reviewed by new translators in order to check other possible mistakes. Thank you for letting us know.

Point 4: In my humble opinion, it could be useful to describe in more detail the practical implications of this research. For instance, why this publication is useful and how could help psychologist?

Response 4: Thank you for your recommendation. We added some new text that explains how a psychologist can be benefited by this study. The new lines:

“Therefore, a Psychology professional can take advantage of this study due to the obtaining of a positive result in the assessment of the BES. With this information, the professional can, among other proposals, perform empirical studies in which the variable empathy is correlated with other variables of interest, measure the basal level of empathy among a sample in which a social intervention can be done, or assess the effects that an intervention has had on the empathy of a population.”

New information included
The authors of the manuscript included some new information in the text and considered it relevant to notify the reviewers of this decision. These are the lines, located in the discussion:
“This argument leads to a practical implication: the dimensionality corroboration and the group equivalence must be assessed in substantive studies, due to the fact that these are not static properties and they are directly related with the reliability estimation [80]. This corroboration can be considered even more strict when the instrument has been derived from a process of adaptation from another cultural context [79].”
“In particular, some of the required aspects advised to be assessed in each study are response patterns such as insufficient effort or neglected answers, expressed as excessive consistency or inconsistency, as well as the appropriate model to measure the answers (congeneric or tau-equivalent [80,81]).”

Reviewer 2 Report

Thank you for the opportunity to review this MS. I will not summarize results for the sake  of brevity. I am very skeptical about its merit for publication. The MS attempts to combine  a narrative review of studies that have utilized the BES combined with a statistical meta-analysis of reliability indices. This is quite novel as usually one will see either the one or the other, not the two combined.

There are concerns, however, regarding the validity of the approach. It is not clear, for example, as to what meta-analytic strategies were employed for extrapolating the observed reliability indices. Moreover, why did authors not go on to meta-analyze the relationships with the various outcome variables studied as well? Why are, in some tables, observations made in % of articles read. This is quite novel to me, I have  never seen such an approach taken in meta-analyses, at least in the mainstream (APA, BPS, European) psychology journals .

Some other comments:

1. The first article in which it was described and validated by its authors has been cited more 109 than 850 times - On which bibliometric dbase is this based?

2. I am not sure as to what kinds of reliable information one can derive from Table 1.

3. Some general information regarding validity of the instrument is provided in section 3.2.4 yet the information is very general and abstract there; for example, it is not explained  the type of validity (construct, discriminant, etc. ) and what how the variables were operationalized

4. What is 'Generalization of the instrument' (Title) and what is '3.3.2. Generalization of reliability '? Please look up  a solid meta-analysis guide in the social sciences. Explain how is reliability expected to be tested meta-analytically (e.g. I did not catch glimpse of correcting for sample size effects). 

The MS is very badly written. It was really difficult to follow through because of  limitations in grammar and using the correct scientific terminology (in the right way).  Please advice with a native English speaker expert in the Social and behavioral sciences.

Some examples, but the MS is full of inaccuracies and grammatic errors that confuse the reader?

Figure 3. Scatterplot of articles’ publication over time.

there is a great international richness,

whom together with other collaborators, has published

which is focused on the stages of child-100 hood.

The BES is one of the most internationally used instruments to measure empathy [12

Figure 2. Progression of publications that include the BES instrument.

Author Response

Response to Reviewer 2 Comments

Reviewer 2: Thank you for the opportunity to review this MS. I will not summarize results for the sake of brevity. I am very skeptical about its merit for publication. The MS attempts to combine  a narrative review of studies that have utilized the BES combined with a statistical meta-analysis of reliability indices. This is quite novel as usually one will see either the one or the other, not the two combined.

There are concerns, however, regarding the validity of the approach. It is not clear, for example, as to what meta-analytic strategies were employed for extrapolating the observed reliability indices. Moreover, why did authors not go on to meta-analyze the relationships with the various outcome variables studied as well? Why are, in some tables, observations made in % of articles read. This is quite novel to me, I have  never seen such an approach taken in meta-analyses, at least in the mainstream (APA, BPS, European) psychology journals.

Authors: We really appreciate your work reviewing this study. First of all, we would like to thank you for all your comments. They are truly considered useful for us in order to grow in our research careers. Regarding the novelty of the approach, we considered it was very helpful to upgrade a common meta-analysis with the insertion of the systematic revision and the validation analysis in order to have an overview of the instrument. We definitely agree with you, it is not common, but at the same time we believe it can be helpful for the readers to have all of this information for them to explore the scale as deeply as possible.

On the other hand, regarding the meta-analytic strategies for extrapolating the observed reliability, we added new text that justifies our approach:

“On the other hand, the results obtained can be generalized when a good degree of confidence can be obtained. That is because, in order to generalize the results to future studies not similar to those of this study, the random coefficients model is generally accepted as the recommended option, and this goal is one of the preferred research goals (Schmidt, Oh, & Hayes, 2009*).”

*Schmidt, F. L., Oh, I.-S., & Hayes, T. L. (2009). Fixed- versus random-effects models in meta-analysis: Model properties and an empirical comparison of differences in results. British Journal of Mathematical and Statistical Psychology, 62, 97-128. doi:10.1348/000711007X255327

Point 1: The first article in which it was described and validated by its authors has been cited more than 850 times - On which bibliometric dbase is this based?

Response 1: Thank you for letting us know that this information was not given in the manuscript, we have added it in the text:

“The first article in which it was described and validated by its authors has been cited more than 850 times, with a progressive increase, according to a search in Google Scholar that the authors themselves carried out in 2018 [12].”

Point 2: I am not sure as to what kinds of reliable information one can derive from Table 1.

Response 2: We added this new paragraph related to the information derived from Table 1:

“For descriptive purposes, Table 1 shows the percentages of articles that provide information related with each of the 5 standards proposed by the guideline “Standards” [31]. An unequal distribution throughout the articles is observed. More detailed information related to the results of each standard is presented in the following sections. This distribution was conducted by the authors of this study, according to the recommendations of the guideline until a consensus was reached. A brief description for the presence of each validity evidence is given in Appendix A.”

However, it is better described in the following sections, where the results of each standard are explained. Thank you very much for this comment, we believe this new paragraph will make it easier for the readers to understand the idea.

Point 3: Some general information regarding validity of the instrument is provided in section 3.2.4 yet the information is very general and abstract there; for example, it is not explained  the type of validity (construct, discriminant, etc.) and what how the variables were operationalized.

Response 3: The constructs used to measure convergent or discriminant validity are described in Appendix A. They were classified according to the information given by each article, as it is explained in the subsection 2.2 Description of the validity study. However, we included new text in this section in order to make it clearer for the reader. Thank you for noticing that. The new text:

“The information that helped to perform this analysis was given by the articles that were chosen, meaning that just the information presented there was considered to do this study.”

Point 4: What is 'Generalization of the instrument' (Title) and what is '3.3.2. Generalization of reliability '? Please look up  a solid meta-analysis guide in the social sciences. Explain how is reliability expected to be tested meta-analytically (e.g.  I did not catch glimpse of correcting for sample size effects). 

Response 4: We are sorry for the confusion that could be generated by the literal translation. We followed the REGEMA checklist, which was designed to assess the reliability generalization of scales through the reported reliability indices. Both headlines, the title of the manuscript and the one in the subsection 3.3.2. are now corrected. We hope it makes it easy for a reader to follow.

Reviewer 2: The MS is very badly written. It was really difficult to follow through because of  limitations in grammar and using the correct scientific terminology (in the right way).  Please advice with a native English speaker expert in the Social and behavioral sciences.

Some examples, but the MS is full of inaccuracies and grammatic errors that confuse the reader?

Figure 3. Scatterplot of articles’ publication over time.

there is a great international richness,

whom together with other collaborators, has published

which is focused on the stages of child-100 hood.

The BES is one of the most internationally used instruments to measure empathy [12

Figure 2. Progression of publications that include the BES instrument.

Authors: Thank you for letting us know all of these inaccuracies, these examples are currently corrected in the manuscript. Also, the whole text is now being reviewed by a native speaker, we hope it makes it easier to read.

New information included
The authors of the manuscript included some new information in the text and considered it relevant to notify the reviewers of this decision. These are the lines, located in the discussion:
“This argument leads to a practical implication: the dimensionality corroboration and the group equivalence must be assessed in substantive studies, due to the fact that these are not static properties and they are directly related with the reliability estimation [80]. This corroboration can be considered even more strict when the instrument has been derived from a process of adaptation from another cultural context [79].”
“In particular, some of the required aspects advised to be assessed in each study are response patterns such as insufficient effort or neglected answers, expressed as excessive consistency or inconsistency, as well as the appropriate model to measure the answers (congeneric or tau-equivalent [80,81]).”

Round 2

Reviewer 1 Report

Congratulations!